# Biosynthetic Gene Content of the ‘Perfume Lichens’ *Evernia prunastri* and *Pseudevernia furfuracea*

**DOI:** 10.3390/molecules24010203

**Published:** 2019-01-08

**Authors:** Anjuli Calchera, Francesco Dal Grande, Helge B. Bode, Imke Schmitt

**Affiliations:** 1Institute of Ecology, Evolution and Diversity, Goethe University Frankfurt, 60438 Frankfurt am Main, Germany; Anjuli.Calchera@senckenberg.de; 2Senckenberg Biodiversity and Climate Research Centre (S-BiKF), Senckenberg Gesellschaft für Naturforschung, 60325 Frankfurt am Main, Germany; Francesco.DalGrande@senckenberg.de; 3Molecular Biotechnology, Department for Biosciences, Goethe University Frankfurt, 60438 Frankfurt am Main, Germany; h.bode@bio.uni-frankfurt.de

**Keywords:** lichen secondary metabolites, tree moss, oakmoss, biosynthetic gene clusters, polyketide synthases, non-ribosomal peptide synthetases, terpene synthases, transcription factor, phylogeny, comparative genomics

## Abstract

Lichen-forming fungi produce a vast number of unique natural products with a wide variety of biological activities and human uses. Although lichens have remarkable potential in natural product research and industry, the molecular mechanisms underlying the biosynthesis of lichen metabolites are poorly understood. Here we use genome mining and comparative genomics to assess biosynthetic gene clusters and their putative regulators in the genomes of two lichen-forming fungi, which have substantial commercial value in the perfume industry, *Evernia prunastri* and *Pseudevernia furfuracea*. We report a total of 80 biosynthetic gene clusters (polyketide synthases (PKS), non-ribosomal peptide synthetases and terpene synthases) in *E. prunastri* and 51 in *P. furfuracea*. We present an in-depth comparison of 11 clusters, which show high homology between the two species. A ketosynthase (KS) phylogeny shows that biosynthetic gene clusters from *E. prunastri* and *P. furfuracea* are widespread across the Fungi. The phylogeny includes 15 genomes of lichenized fungi and all fungal PKSs with known functions from the MIBiG database. Phylogenetically closely related KS domains predict not only similar PKS architecture but also similar cluster architecture. Our study highlights the untapped biosynthetic richness of lichen-forming fungi, provides new insights into lichen biosynthetic pathways and facilitates heterologous expression of lichen biosynthetic gene clusters.

## 1. Introduction

Lichens are symbioses, composed of a fungal partner (mycobiont) and one or more photosynthetic partners (photobiont) [1]. Approximately one fifth of all described fungi form lichens and more than 19,000 lichen species are described [2]. Lichens can be found in most terrestrial ecosystems. They impact various community processes, also because of their rich and diverse secondary chemistry [3,4]. Recent studies have shown that lichens represent complex multi-species symbiotic assemblages, forming microhabitats and harbouring a high diversity of other eukaryotic and prokaryotic microorganisms [5,6,7]. High-throughput sequencing technologies have revealed the presence of multiple fungal and algal species, along with hyper-diverse bacterial communities, within individual lichen thalli [7,8,9,10,11,12,13].

Lichens produce a great variety of natural compounds, and more than 1000 secondary metabolites have been identified to date, many of which are exclusively found in lichens [14,15,16,17]. The vast majority of these characteristic lichen secondary metabolites is of fungal origin. Many lichen secondary metabolites have important ecological roles including light-screening, chemical weathering, allelopathic and anti-herbivore defence [18,19,20]. Furthermore, lichens are a promising source for pharmaceutically interesting natural products, because of the manifold biological activities of lichen compounds—including antiviral, antibiotic, antitumor, allergenic, plant growth inhibitory, antiherbivore and enzyme inhibitory activities [15,19,21,22]. For example, the secondary metabolites atranorin, evernic acid, physodic acid and usnic acid found in the lichens *Evernia prunastri* and *Pseudevernia prunastri* are strong metabolic enzyme inhibitors, and atranorin may inhibit lung cancer cell mobility and tumorigenesis [22,23]. A wide range of lichens have been used in traditional medicines all around the world [24]. Today, one of the more significant economic uses of lichens is in the perfume industry. Combined lichen material is used for extracts added to perfumes for a sweet and mossy smell and to ensure persistence on the skin. Several hundred to thousands of tons of *E. prunastri* and *P. furfuracea* are harvested every year in France, Morocco and South-eastern Europe for this purpose [16,19,25,26].

The most abundant classes of lichen secondary metabolites are phenolic compounds, dibenzofurans, depsides and depsidones built by multienzyme polyketide synthases (PKS). Derived monoaromatic subunits (orcinol, β-orcinol type or methylphloroacetophenone) are then linked by ester, ether or carbon-carbon bonds [1,22,27]. Fungal PKSs consist of a set of active site domains used in an iterative fashion for multiple catalytic cycles and are subdivided into non-reducing (NR-PKS), or reducing (R-PKS) according to the level of reductive processing [28]. Both types of polyketides are found in lichen-forming fungi but fully oxidized polyketides account for most of the well-known and common lichen substance classes such as depsides and depsidones whose formation is controlled by NR-PKS genes [16,29]. Fungal secondary metabolic pathway genes are often grouped in gene clusters including genes encoding tailoring functions, transporters, and pathway-specific regulatory genes along with the core genes [30,31,32].

Recent advances in genome sequencing and bioinformatic mining algorithms have enabled the identification and characterization of biosynthetic gene clusters from genome sequences and led to the discovery of thousands of biosynthetic gene clusters [33,34,35,36]. However, the identity, structure and function of the metabolites encoded by these gene clusters remain mostly unknown. The clusters are therefore referred to as cryptic or orphan clusters [37]. In particular, fungal PKS gene clusters are challenging to functionally characterize, mainly due to the unpredictable intrinsic cryptic programming concealed within iterative PKS genes themselves and missing knowledge on precise selectivity and activity of tailoring enzymes [30,31,38]. From the vast biosynthetic potential found in genomes only a small fraction has been investigated experimentally to connect genes to metabolite production—for example through gene knockout or heterologous expression. But the presence of cryptic or orphan gene clusters in many genomes, and the fact that a vast number of biosynthetic genes is not reflected by the metabolic profile, hint at a plethora of yet undiscovered chemical compounds [32,39,40].

Despite the rich chemical diversity found in lichens, only few biosynthetic studies are available. In lichen-forming fungi the slow growth rates and difficulties in aposymbiotic mycobiont cultivation further hinder the characterization of the molecular mechanisms underlying the biosynthesis of characteristic lichen metabolites [14,41]. The biosynthesis of substances may depend on abiotic factors such as geographic, altitudinal or microhabitat conditions or the response to microclimatic fluctuations, seasonality, chemical signals, hydration, or UV radiation [14,22]. Biosynthesis of natural products may also be influenced by biological factors, such as the presence of competing plant or lichen species, predation by insects, or contact to other symbionts that are part of the lichen [27]. Furthermore, lichen mycobionts synthesize significant quantities of secondary metabolites only under permissive conditions and therefore the production in axenic cultures can differ substantially from that in nature [21,22,42].

In lichen-forming fungi no secondary metabolite has been directly linked with experimental proof (i.e., gene knockout, heterologous expression) to a biosynthetic gene cluster, although several assignments have been proposed. The first identification of a lichen secondary metabolic gene cluster—grayanic acid in *Cladonia grayi*—was accomplished by Armaleo et al. [43] combining phylogenetic analysis, biosynthetic prediction from PKS domain arrangement and correlation of mRNA levels with metabolic production. Later, the putative gene clusters encoding usnic acid and 6-hydroxymellein in *Cladonia uncials* were identified by Abdel-Hameed et al. [41,44] on the basis of PKS domain arrangement and prediction of the functions of tailoring enzymes encoded in the same gene cluster.

Several methods are applied and combined to discover PKS genes in lichenized fungi: amplification of PKS gene fragments from genomic DNA using degenerate primers and design of probes for library screening; the use of cDNA-based templates generated from reverse-transcribed mRNA, or gaining biosynthetic insights directly from transcriptomes or genomes sequenced from mycobiont cultures or reconstructed from metagenomic thalli [45,46,47]. The latter became possible through the advances of sequencing technologies and taxonomic assignment tools and led to the publication of several draft genomes of lichen-forming fungi over the last few years (see below). For the prediction of biosynthetic function of PKS genes there are diverse strategies, apart from experimental evidence through gene knockout, heterologous expression, radiolabelling or oxidation experiments [48,49,50,51]. These include phylogenetic approaches to infer the functions through a close phylogenetic relatedness to characterized genes [52,53], transcriptional profiling [43,47], or comparative homology mapping of entire gene clusters [31,36,54]. These approaches are often combined to provide putative assignments [47]. A review on reported PKS genes from lichenized fungi including an overview on approaches for PKS identification and characterization has been published recently and highlights the prospects of genomics-driven natural product discovery in lichens [45].

Here, we combine genome mining with comparative genetic mapping and phylogenetics of the two lichenized fungi *Evernia prunastri* and *Pseudevernia furfuracea*. Both lichens are of economic importance [1,19] and their fungal genomes contain a high number of biosynthetic gene clusters [46]. Moreover, a comprehensive review on the multitude of extracts identified from both lichens is available [25,26] and includes, for example, the bioactive metabolite atranorin detected in both lichens. The secondary metabolite richness and partially overlapping chemical profiles make these species particularly interesting study systems for combining genome mining and comparative genomics to investigate biosynthetic enzymes that may be responsible for the production of characteristic lichen substance classes. We further compare the biosynthetic gene richness found in these species with that of 13 other lichenized fungi and 57 representative fungal species and thereby provide the most complete comparison of biosynthetic gene families in lichen genomes based on entire genomes. We present mapped biosynthetic gene clusters of *E. prunastri* and *P. furfuracea* with functionally annotated accessory genes and putative regulatory genes. These carefully annotated clusters provide baseline information for heterologous expression of lichen biosynthetic gene clusters. Furthermore, we present all PKS genes mined from 15 genomes of lichenized fungi in a phylogenetic framework of characterized biosynthetic genes. Information on phylogenetic relatedness to previously characterized genes broadens our understanding of lichen biosynthetic pathways and may help us to identify promising clusters for the production of characteristic substance classes.

Specifically, we address the following questions:(I)What is the diversity of biosynthetic gene clusters in *Evernia prunastri* and *Pseudevernia furfuracea* and how does it compare to other lichenized fungi and non-lichenized fungi?(II)What is the architecture and gene content of those clusters with high homology between *E. prunastri* and *P. furfuracea*?(III)Where do PKSs from *E. prunastri* and *P. furfuracea* group phylogenetically in a phylogeny of PKSs with known functions?

## 2. Results & Discussion

### 2.1. Biosynthetic Gene Richness in Fifteen Annotated Genomes of Lichen-Forming Fungi

We investigated the biosynthetic gene richness in a total of fifteen genomes of lichen-forming fungi. The species belong to different phylogenetic groups and synthesize a variety of lichen substances (Table 1).

The genomes differ substantially in assembly status—from 7 to 3891 scaffolds—but most are 90–96% complete according to BUSCO (Figure 1a). Only the genomes of *Ramalina peruviana* (86%) and *Cladonia uncialis* (89%) score lower in completeness and also have the lowest scaffold N50 (see genome statistics given in Table 1 and Appendix A). Since seven genomes did not have available gene sets, we predicted gene models. The limitations of gene prediction without transcriptome data are evident in the assessment of gene set completeness in Figure 1b. The gene sets for which we performed gene annotation are less complete (72–80% compared to 82–97%) and show more fragmented BUSCO marker genes. Nevertheless, the predicted number of genes is in the range of most of the gene sets available for lichen-forming fungi (~8200 to 11,800). The only exception here is the genome of *Lobaria pulmonaria* which has a considerable larger genome size with 56.1 Mb and 15,607 genes but also the highest rate of duplicated marker genes (Table 1, Appendix A). Even though the gene prediction done here is not optimal, it improves the downstream biosynthetic gene cluster detection over the less specific annotation that can be done within antiSMASH [62].

We identified a high number of biosynthetic gene clusters in all genomes of lichen-forming fungi (Table 2). The fungal version of antiSMASH annotated on average ~47 clusters per genome, ranging from 27 clusters for *Umbilicaria pustulata* up to as many as 80 gene clusters in *Evernia prunastri*. The most abundant family of secondary metabolite enzymes identified in all genomes are reducing type I polyketides (R-PKS) followed by terpene synthases and non-reducing type I polyketides (NR-PKS). We also detected non-ribosomal peptide synthetases (NRPS), hybrid PKS-NRPS and type III PKSs in most genomes.

Polyketides represent the most abundant class of lichen secondary metabolites and can be reduced or fully oxidized [16,22,27]. It has been reported that most polyketides found in lichens are fully oxidized [63]. Nevertheless, we find that the number of R-PKS genes exceeds by far the number of NR-PKS genes. The genomes of the lichen-forming fungi show—through their large number of secondary metabolic genes and gene clusters—the potential for a much greater number of natural products than have been reported to occur in the respective lichen species (Table 1).

The genomes of lichen-forming fungi in the Lecanoromycetes show a remarkable richness of secondary metabolic gene clusters in comparison with genomes of species from all major fungal classes. We present the number of predicted secondary metabolite gene clusters in 15 genomes of lichenized fungi (this study) and 57 ascomycete and basidiomycete genomes of non-lichenized fungi analysed in a previous study [32] (Figure 2). The average number of predicted secondary gene clusters is ~31 clusters and the average number of predicted PKS clusters is ~11. Except for two *Umbilicaria* species the number of lichen gene clusters is above average and all lichen genomes harbour more than the average amount of PKS gene cluster (~24 clusters on overage). Indeed, the highest number of PKS gene clusters are found in *Evernia prunastri* (38 PKS clusters) together with six other lichens including *Pseudevernia furfuracea* in the top ten PKS cluster count. We used a non-representative selection of fungal genomes for the comparison and thus did not include all fungal species that are reported to be rich in secondary metabolite gene clusters, for example, *Pestalotiopsis fici* (Sordariomycetes) [64]. The comparison is meant for the sole purpose of placing the richness of secondary metabolite gene clusters found in lichen-forming fungi in a broader context (Figure 2).

### 2.2. Gene Cluster Comparison

*Pseudevernia furfuracea* and especially *E. prunastri* show a high number of biosynthetic gene clusters (Table 2), and a high number of natural products has been reported for both species [25,26]. This richness in secondary metabolites and partially overlapping chemical profiles (e.g., atranorin and chloroatranorin) make these species particularly interesting study systems for combining genome mining and comparative genomics to investigate biosynthetic enzymes that may be responsible for the production of characteristic lichen substance classes. Moreover, both species are harvested in large quantities for the perfume industry and are of economic importance [1,19]. A detailed list of the biosynthetic gene clusters detected in both species can be found in Appendix A.

For the gene cluster comparison, we functionally annotated 887 cluster genes of *E. prunastri* and 548 cluster genes of *P. furfuracea* with gene ontology terms and protein names. The full Blast2GO annotation is presented in Appendix A. We then identified 126 orthologous pairs as Reciprocal Best Blast Hits (RBH) between the *E. prunastri* and *P. furfuracea* cluster genes. The full list of RBHs is given in Appendix A. From these results we chose all biosynthetic gene clusters that contained an orthologous core PKS, NRPS or hybrid PKS-NRPS for our comparative genetic mapping approach presented in the following synteny plots of Figure 3, Figure 4 and Figure 5. The phylogenetic placements of these clusters containing a ketosynthase (KS) domain are presented below.

Four NR-PKS clusters show a core RBH gene (Figure 3). Three of these (Figure 3A,B,D) show a high cluster homology between *E. prunastri* and *P. furfuracea* with several genes carrying similar functional annotation and RBHs of each other. The homologous genes include for example cytochrome P450, monooxygenases, serine/threonine kinases, and regulatory genes. Most of the homologous regions between the clusters are confined within genes but there are a few exceptions (Figure 3D). These exceptions with high homology of coding sequences to non-coding regions may be an artefact of gene prediction, and may indicate genes missed in the annotation. The clusters in Figure 3C on the other hand display only similarity between the two core PKS genes—one of which is a R-PKSs—and not between any accessory genes. Closer inspection of the domain architecture of the NR-PKSs in Figure 3 using BLASTp indicates the presence of acyl carrier protein (ACP) transacylase starter units (SAT) and product templates (PT). Both conserved domains were shown to be typical for NR-PKSs in addition to the minimal PKS domain structure of a KS, AT (acyltransferase) and ACP [65,66,67,68].

We further identified four R-PKS clusters with orthologous core genes (Figure 4). Two cluster comparisons (Figure 4A,B) show a high homology between both species. The entire R-PKS in Figure 4A is split over two to three genes—possibly an artefact of gene prediction—but we detected nine RBH accessory genes in the clusters including oxidases and calcium-binding domain proteins. The homologous cluster genes in Figure 4B include cytochrome P450, dehydrogenases, transporter genes and chalcone synthases (type III PKS). The other two comparative mapped clusters (Figure 4C,D) share only the core gene as a RBH. In Figure 4C we detected a homologous region with annotated genes in *P. furfuracea* while *E. prunastri* lacked any predicted gene models in the concurrent region.

Last, we included NRPS and hybrid PKS-NRPS clusters with orthologous core genes in the comparative analyses (Figure 5). We detected two hybrid PKS-NRPS clusters (Figure 5A,C) and two NRPS clusters (Figure 5B,C) all showing a high similarity in functionally annotated genes and through RBHs. Accessory genes include transporter genes, hydrolases and putative regulatory genes. In Figure 5C a hybrid PKS-NRPS and a NRPS are encoded closely together. This might suggest that both enzymes participate in the biosynthesis of one natural product or that the two natural products might function together.

### 2.3. Putative Regulators of Biosynthetic Gene Clusters

We identified a total of 60 putative intracluster regulators in the biosynthetic gene clusters of *E. prunastri* and *P. furfuracea*. Most of these were present in the gene clusters of *E. prunastri* (44 genes). Two pairs are orthologs between both species based on RBH analysis and can be found among the clusters investigated with comparative mapping (see clusters in Figure 3B and Figure 5C). We detected ten C6 zinc finger domain proteins, six Zn_2_Cys_6_ DNA-binding proteins and two ankyrin repeat proteins among other fungal specific transcription factors. These are typical regulators of secondary metabolites [37,69]. A detailed list of these putative regulatory genes can be found in Appendix A.

### 2.4. Phylogenetic Analysis with Characterized Fungal Polyketides

We reconstructed a maximum likelihood tree of the conserved KS domain that allows inference of domain architecture and pathway association of PKSs [52,70,71]. We included a total of 413 KS sequences from fifteen genomes of lichen-forming fungi in relation to 131 MIBiG entries of characterized fungal PKSs and six animal fatty acid synthases (FAS) as an outgroup. We also included partial PKSs from lichen-forming fungi that lack one of the required domains AT or ACP for a minimal PKS organization. Out of 413 lichen KS sequences five originate from a PKS without an AT domain and 136 from a PKS without an ACP domain. We included these since such partial PKSs are also present in the MIBiG repository of characterized enzymes [72] and may nevertheless represent functional and interesting genes. In our manually curated MIBiG dataset we found one entry without an AT domain and 32 entries without an ACP domain.

The entire phylogenetic tree is shown in full length in Appendix A. Overall the phylogeny shows support for a clade with NR-PKSs, a clade with R-PKS and a clade containing 6-methylsalicyclic acid synthases (6-MSAS). Hybrid PKS-NRPS genes contain reducing domains and fall within the clade of R-PKS. We provide here a few examples of interesting clades found in the phylogenetic analysis with a focus on NR-PKS since the basic building subunits for lichen-specific depsides and depsidones are believed to be encoded by this gene family [1,14,27,73].

Within the NR-PKS clade we find one supported clade of PKSs containing C-methyltransferase (cMT) domains including a subclade with the PKS proposed to produce the lichen substance usnic acid [41]. *Evernia prunastri* is also known to produce usnic acid (Table 1) and one PKS (gene 02873) falls phylogenetically close to the potential usnic acid producer (Figure 6).

Another clade contains the NR-PKSs of *E. prunastri* and *P. furfuracea* (Figure 3A) which are phylogenetically close to the PKS gene that is putatively associated with grayanic acid biosynthesis [43]. All *Cladonia* species included in this study (*C. grayi*, *C. macilenta*, *C. metacorallifera* and *C. uncialis*) have members within this clade (Figure 7).

We further inspected the phylogenetic placement of the comparative mapped NR-PKSs of the *E. prunastri* and *P. furfuracea* gene clusters. The gene clusters mapped in Figure 3B fall into a group with tandem ACP domains and in the proximity of characterized PKSs involved in the production of hydroxy naphthalenes [74,75] (Figure 8). Naphthalene is found in the extracts of *E. prunastri* and *P. furfuracea* used for the perfume industry [1] (pp. 132–133).

Two closely related clusters of *E. prunastri* and *P. furfuracea* (Figure 3C) fall into a clade with the experimentally characterized PKS gene *orsA* known to be involved in orsellinic acid biosynthesis in *Aspergillus nidulans* [76]. This clade (Figure 9) provides an especially interesting starting point for further investigations as the most abundant class of lichen metabolites are composed of orcinol or β-orcinol monomers [27].

We also want to underline that combining comparative mapping of entire gene clusters with a phylogenetic approach based only on the KS sequence of the core PKS gene shows that entire cluster similarities correspond to KS topology. This is in line with findings that emphasize the predictive power of the conserved KS domains for the investigation of enzyme architecture and pathway association [52,70,71].

Our study combines genome mining and comparative genomics and highlights the high diversity of biosynthetic gene clusters that can be found in fifteen genomes of lichen-forming fungi. This number exceeds by far the number of lichen metabolites that are reported for these species. The secondary metabolite gene cluster richness found in the genomes of lichen-forming fungi is above the average richness found in other representative fungal species, especially polyketide synthase gene clusters. The comparative mapping of interesting biosynthetic gene clusters, functional annotation of accessory genes together with the identification of putative regulatory genes presented here will aid in providing new insights into lichen biosynthetic pathways and serve as a valuable resource for developing heterologous expression of lichen biosynthetic gene clusters.

## 3. Materials and Methods

### 3.1. Identification and Annotations of Biosynthetic Gene Clusters

We used the genomes of the two lichen-forming fungi of *Evernia prunastri* and *Pseudevernia furfuracea* for biosynthetic gene cluster identification (accession numbers in Table 1). Sequencing of cultures, genome assemblies, gene prediction and genome mining for biosynthetic gene clusters in both species were done as described in Meiser et al. [46]. We provide here a short description on the identification of biosynthetic gene clusters to give a full depiction on our focused genome mining approach. We identified typical gene families of secondary metabolism and annotated biosynthetic gene clusters with the fungal version of antiSMASH v.4.0.2 (fungiSMASH) [62,77], including polyketide synthases (PKS), non-ribosomal peptide synthetases (NRPS) and terpene synthases. These gene families are typical targets in lichen-metabolite genome mining, because they encode the structural frame of most secondary metabolites [19,37,41,44]. As input for antiSMASH we used an annotated nucleotide file (EMBL format) constructed from the genome FASTA file and the GFF file from gene prediction.

Both investigated species show a high number of biosynthetic gene clusters with reducing and non-reducing PKSs, NRPSs, and terpene synthases [46] and represent a promising source of bioactive secondary metabolites. To further investigate these biosynthetic gene clusters for accessory genes involved in the lichen metabolite synthesis, we annotated all predicted protein sequences belonging to an antiSMASH cluster with gene ontology (GO) terms and protein names. The functional annotation was done using Blast2GO [78] v.5.0.22 and by running BLASTp [79] v.2.2.29 against the NCBI GenBank protein database *nr* (downloaded May 2017) [80] and InterProScan [81] v.5.28–67.0 with a matching lookup service and the Pfam (protein family) database v.31.0 [82].

### 3.2. Gene Cluster Comparison in Evernia prunastri and Pseudevernia furfuracea

We applied a two-step approach to compare the biosynthetic genes cluster of *E. prunastri* and *P. furfuracea*. First, as a simple and fast way of detecting orthologous pairs [83], we identified Reciprocal Best Blast Hits (RBH) between all cluster genes of both species. For better detecting orthologs we used BLASTp v.2.2.31 + with final Smith-Waterman alignment and soft filtering (BLAST flags *use_sw_tback*, *soft_masking true*, *seq yes* and *evalue 1e-6*) [83,84]. The BLAST hits were then filtered for a minimum query coverage of 50% (*qcovhsp*) and a minimum identity of 70% over the alignment length (*pident*) and sorted for highest bit-score and lowest e-value [85].

Second, we used comparative genetic mapping to analyse the homology and arrangement of entire clusters instead of looking at core gene orthologs only. For the comparison we chose all biosynthetic gene clusters that contained a RBH core PKS, NRPS, or hybrid PKS-NRPS. Synteny plots were generated with Easyfig python script v.2.2.3 [86] and the tBLASTx option with a minimum identity value of 40% and a minimum alignment length of 50 [87]. Gene clusters were reversed when necessary to have a matching orientation. We generated GBK files with seqkit [88] v.0.7.2 and the seqret tool in the EMBOSS package [89] v6.6.0.0 as input for Easyfig.

### 3.3. Putative Regulators of Biosynthetic Gene Clusters

The transcriptional control of biosynthetic genes by transcription factors is complex and can occur on several levels from pathway-specific to a broader global regulation [37]. Specific acting regulatory genes are usually found in the cluster that the factors regulate. Examples include Zn(II)_2_Cys_6_, Cys_2_His_2_, basic leucine zipper (bZip), winged helix, zinc-finger, or ankyrin repeat proteins [30,69]. The identification of these regulators involved in the synthesis of secondary metabolites may be crucial for successful heterologous expression experiments or for activating silent gene clusters.

For the detection of putative regulators within the biosynthetic gene clusters of *E. prunastri* and *P. furfuracea* we made use of the Blast2GO annotation. We searched for the following key terms in the GO names: ‘signal transducer activity’, ‘transcription factor activity’ and ‘transcription factor binding’.

### 3.4. Biosynthetic Gene Richness in Other Genomes of Lichenized and Non-Lichenized Fungi

We assessed the chemical diversity and biosynthetic gene richness of *E. prunastri* and *P. furfuracea* in comparison to the genomes of other lichen-forming fungi. We analysed all genomes of lichen-forming fungi from fungal culture available in NCBI (National Center for Biotechnology Information) and for authorized genomes in JGI (DOE Joint Genome Institute) (Table 1). The only genome not derived from fungal culture is the genome of *Umbilicaria pustulata* where the fungal genome was reconstructed with metagenomic sequencing of the lichen thallus instead [90]. We ran antiSMASH as described above on all thirteen additional genomes.

For seven of these genomes no gene set was available and gene models had to be predicted before running antiSMASH to improve the biosynthetic gene cluster detection. The de novo gene prediction and annotation of these genomes was done with MAKER [91] v2.31.8 using an iterative approach as recommended in the protocols of Campbell et al. [92]. In the first round of MAKER we used Hidden Markov Models (HMMs) generated with GeneMark-ES [93] v4.33 and SNAP [94] with hints from CEGMA [95] v2.4 (performed on iPlant [96]) as training data for gene finding. The first-round results were then converted to new SNAP and Augustus [97] v3.0.2 HMMs for the second round of MAKER resulting in the final set of gene models.

The genome and gene set completeness of all 15 lichen genomes was assessed based on evolutionarily-informed expectations of gene content with BUSCO (Benchmarking Universal Single-Copy Orthologs) [98] v.3.0 and a lineage-specific set of 1315 Ascomycota single-copy orthologs from OrthoDB [99] v.9. Completeness estimates were visualized with the package ggplot2 [100] v.2.2.1 in R [101] v.3.4.4.

Additionally, we provide an overview on how richness of biosynthetic gene clusters found in lichenized fungi compare to the cluster richness found in non-lichenized fungi. Therefore, we combined the secondary metabolic gene cluster counts for the 15 genomes of lichenized fungi obtained from antiSMASH in this study with a recently published analysis from 57 genomes of representative fungal ascomycete and basidiomycete species [32].

### 3.5. Phylogenetic Analysis with Characterized Fungal Polyketides

We used phylogenetics to analyse the core enzymes of PKS and hybrid PKS-NRPS gene clusters in relation to characterized fungal biosynthetic gene clusters. Combining genome mining with phylogenetic analysis may provide clues for identifying promising clusters for the production of characteristic substance classes through phylogenetic relatedness to previously characterized genes [36,62,102].

Our phylogenetic approach is based on protein sequences of the ketosynthase (KS) domain of PKSs and hybrid PKS-NRPSs. The KS domains are the most conserved and essential part of a PKS cluster and are highly predictive of enzyme architecture and pathway association [52,70,71]. We included all KS sequences identified in the fifteen genomes of lichen-forming fungi (Table 2) that stem from genes with at least three PKS domains and with KS sequences longer than 200 bases. This resulted in a dataset of 413 KS sequences originating from lichen-forming fungi. Additionally, we made use of the MIBiG repository (Minimum Information about Biosynthetic Gene cluster [103]) to include experimentally characterized biosynthetic gene clusters [72]. We compiled a custom database with 131 entries of all fungal PKS records in MIBiG (downloaded January 2018). Further we included six animal fatty acid synthase (FAS) protein sequences as outgroup for the phylogenetic inference (NCBI Reference Sequence accessions: *Bos taurus* NP_001012687.1; *Gallus gallus* NP_990486.2; *Homo sapiens* NP_004095.4; *Mus musculus* NP_032014.3; *Rattus norvegicus* NP_059028.1; *Sus scrofa* NP_001093400.1).

We ran MAFFT [104,105] v7.309 in Geneious v9.1.8 (https://www.geneious.com) to generate a multiple sequence alignment of all 550 KS amino acid sequences resulting in an alignment with a length of 1043 characters. We chose the iterative refinement algorithm G-INS-i with a BLOSUM62 scoring matrix, a gap open penalty of 1.53 and an offset value of 0.123. We performed ProtTest [106] v.3.4.2 on our alignment to select the best-fitting substitution model of amino acid replacement for subsequent tree reconstruction. Model-testing indicated LG + I + G as the best model for both the Akaike and the Bayesian Information Criterion. The phylogenetic tree was then generated with RAxML-HPC Black Box v.8.2.10 (Randomized Axelerated Maximum Likelihood) [107] on CIPRES v3.3 (Cyperinfrastructure for Phylogenetic Research) [108,109] with automated bootstrapping, protein sequence type, FAS sequences set as outgroup, estimating proportion of invariable sites (GTRGAMMA + I), protein substitution matrix LG, no empirical base frequencies and printing branch lengths. The tree was visualized with Figtree v.1.4.3 (http://tree.bio.ed.ac.uk/software/figtree) rooting the tree with the help of the FAS outgroup and ordering nodes increasingly.

## Figures and Tables

**Figure 1 molecules-24-00203-f001:**
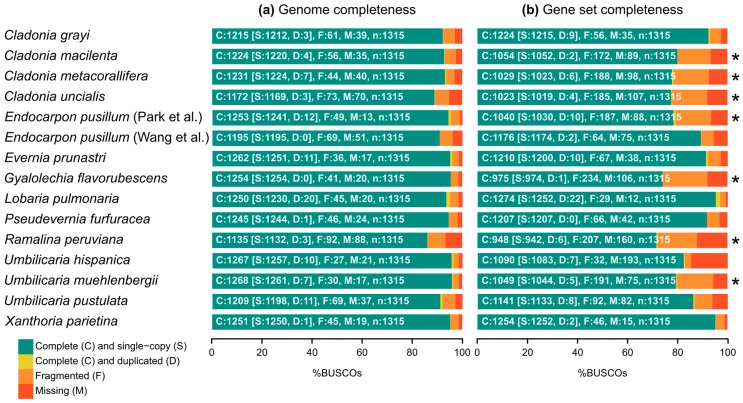
(**a**) Genome completeness for assemblies and (**b**) gene set completeness for gene annotations of lichen-forming fungi. The completeness is compared against 1315 orthologous BUSCO marker genes for Ascomycota. Exact percentages can be found together with basic genome statistics in Appendix A. Asterisk indicates genomes annotated in the present study (without RNA evidence).

**Figure 2 molecules-24-00203-f002:**
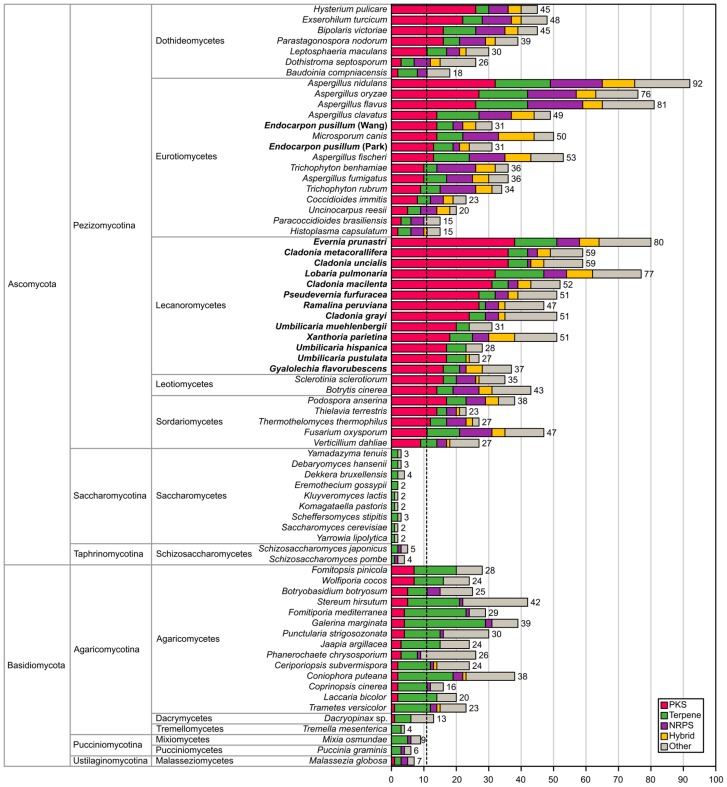
Overview of predicted secondary metabolic gene clusters across the genomes of representative fungal species adopted from [32]. Bold font indicates genomes of lichen-forming fungi included from this study. The dashed line shows the average number of PKS gene clusters found in a genome. “Hybrid” refers to clusters with multiple core genes belonging to different secondary metabolite families. PKS = polyketide synthase; NRPS = non-ribosomal peptide synthetase.

**Figure 3 molecules-24-00203-f003:**
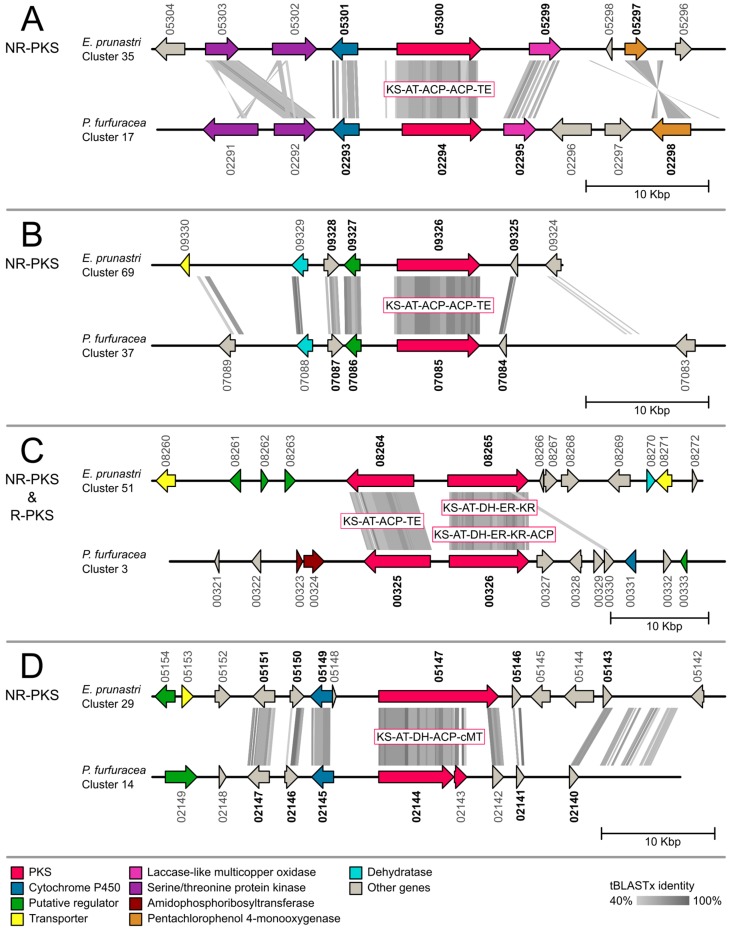
Synteny plots of biosynthetic gene clusters with an ortholog non-reducing polyketide synthase (NR-PKS) core gene in *Evernia prunastri* and *Pseudevernia furfuracea*. Orthologous genes identified with the reciprocal best hit (RBH) approach are highlighted in bold. ACP = Acyl carrier protein; AT = Acyltransferase; cMT = C-Methyltransferase; DH = Dehydratase; ER = Enoylreductase; KR = Ketoreductase; KS = Ketosynthase; TE = Thioesterase.

**Figure 4 molecules-24-00203-f004:**
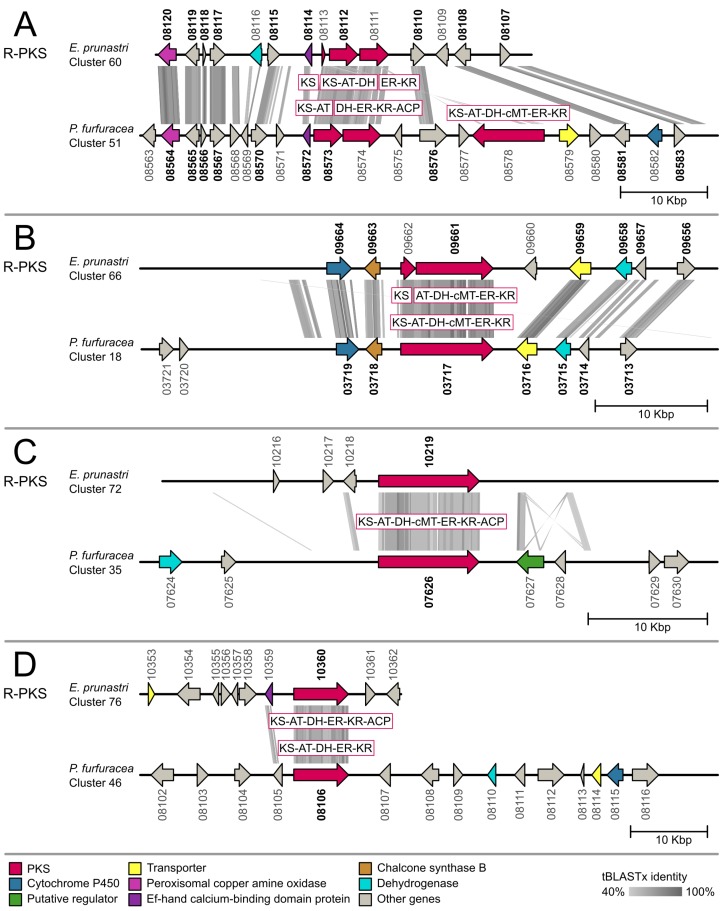
Synteny plots of biosynthetic gene clusters with an ortholog reducing polyketide synthase (R-PKS) core gene in *Evernia prunastri* and *Pseudevernia furfuracea*. Orthologous genes identified with the reciprocal best hit (RBH) approach are highlighted in bold. ACP = Acyl carrier protein; AT = Acyltransferase; cMT = C-Methyltransferase; DH = Dehydratase; ER = Enoylreductase; KR = Ketoreductase; KS = Ketosynthase; TE = Thioesterase.

**Figure 5 molecules-24-00203-f005:**
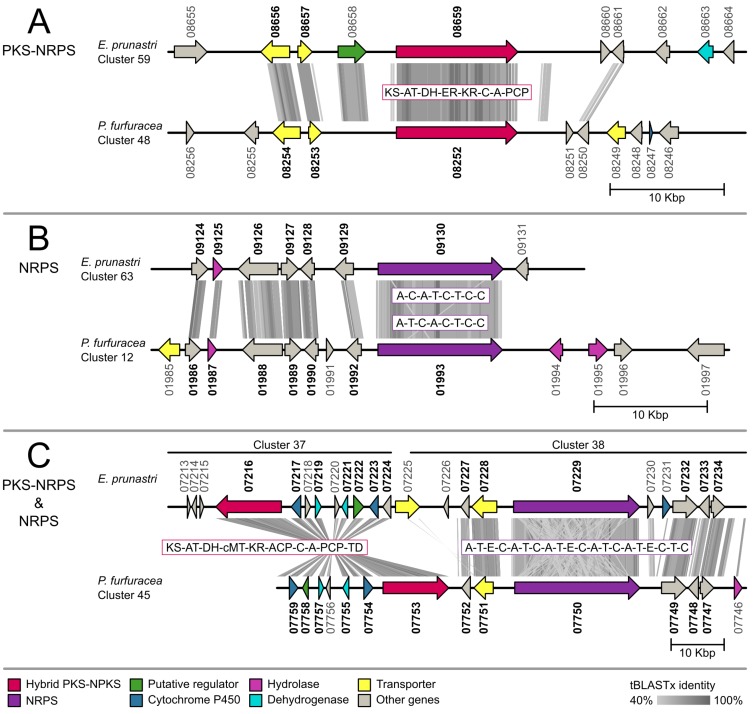
Synteny plots of biosynthetic gene clusters with an ortholog non-ribosomal peptide synthetase (NRPS) core gene or hybrid with polyketide synthase (PKS-NRPS) in *Evernia prunastri* and *Pseudevernia furfuracea*. Orthologous genes identified with the reciprocal best hit (RBH) approach are highlighted in bold. A = Adenylation; ACP = Acyl carrier protein; AT = Acyltransferase; C = Condensation; cMT = C-Methyltransferase; DH = Dehydratase; E = Epimerization; ER = Enoylreductase; KR = Ketoreductase; KS = Ketosynthase; PCP = peptide carrier protein; T = Thiolation; TD = Terminal domain.

**Figure 6 molecules-24-00203-f006:**
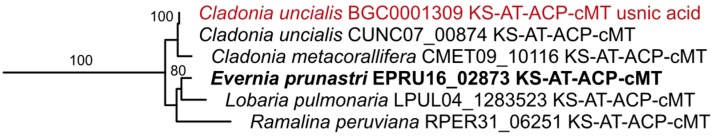
Supported clade containing the gene *MPAS* (methylphloroacetophenone synthase) of *Cladonia uncialis* (MIBiG-ID BGC0001309) putatively associated with usnic acid biosynthesis. For the complete KS tree see Appendix A.

**Figure 7 molecules-24-00203-f007:**
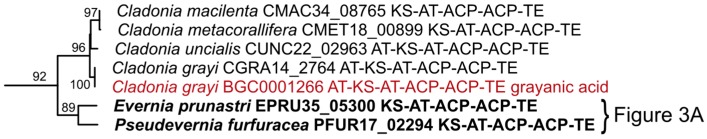
Supported clade containing the gene *PKS16* of *Cladonia grayi* (MIBiG-ID BGC0001266), putatively involved in grayanic acid biosynthesis. The cluster 35 of *Evernia prunastri* (EPRU35) and cluster 17 of *Pseudevernia prunastri* (PFUR17) are phylogenetically close and are presented in detail in synteny plot Figure 3A. For the complete KS tree see Appendix A.

**Figure 8 molecules-24-00203-f008:**
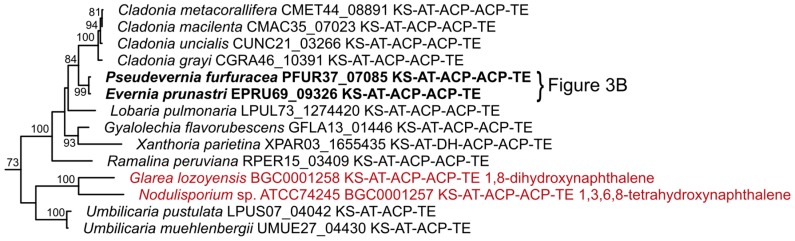
Supported clade containing PKSs from dihydroxy and tetrahydroxy naphthalene producers (MIBiG-ID BGC0001257, BGC0001258). For the complete KS tree see Appendix A.

**Figure 9 molecules-24-00203-f009:**
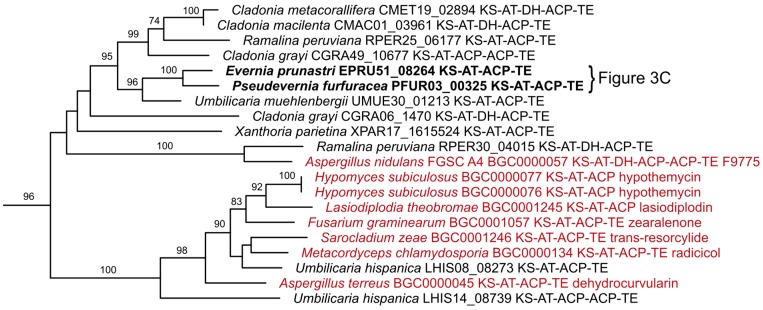
Supported clade containing the PKS *orsA* for orsellinic acid in *Aspergillus nidulans* (MIBiG-ID BGC0000057). For the complete KS tree see Appendix A.

**Table 1 molecules-24-00203-t001:** Overview of genomes of lichen-forming fungi used in this study.

Species	Taxonomic Group	Data Repository ^1^	Gene Set Previously Available	Genome Size	Number of Scaffolds	Scaffold N50	Number of Genes	Abbreviation	Metabolites Reported [55,56,57,58,59]
*Cladonia grayi*	Lecanoromycetes, Lecanorales	JGI Clagr3 v2.0	yes	34.6 Mb	414	243,412	11,389	CGRA	grayanic acid, fumarprotocetraric acid complex
*Cladonia macilenta*	Lecanoromycetes, Lecanorales	NCBI AUPP00000000.1	no	37.1 Mb	240	1,469,036	10,559	CMAC	thamnolic acid, barbatic acid, didymic acid, squamatic acid, usnic acid, rhodocladonic acid
*Cladonia metacorallifera*	Lecanoromycetes, Lecanorales	NCBI AXCT00000000.2	no	36.7 Mb	30	1,591,850	10,497	CMET	usnic acid, didymic acid, squamatic acid, rhodocladonic acid
*Cladonia uncialis*	Lecanoromycetes, Lecanorales	NCBI NAPT00000000.1	no	32.9 Mb	2124	34,871	10,902	CUNC	usnic acid, squamatic acid
*Endocarpon pusillum* (Park et al.) [60]	Eurotiomycetes, Verrucariales	NCBI JFDM00000000.1	no	37.2 Mb	40	1,340,794	11,756	EPUP	(none reported)
*Endocarpon pusillum* (Wang et al.) [61]	Eurotiomycetes, Verrucariales	NCBI APWS00000000.1	yes	37.1 Mb	908	178,225	9238	EPUW	(none reported)
*Evernia prunastri*	Lecanoromycetes, Lecanorales	NCBI NKYR00000000.1	yes	40.3 Mb	277	264,454	10,992	EPRU	evernic acid, atranorin, usnic acid
*Gyalolechia flavorubescens*	Lecanoromycetes, Teloschistales	NCBI AUPK00000000.1	no	34.5 Mb	36	1,693,300	10,460	GFLA	parietin, emodin, fallacinal, fragilin
*Lobaria pulmonaria*	Lecanoromycetes, Peltigerales	JGI Lobpul1 v1.0	yes	56.1 Mb	1911	50,541	15,607	LPUL	stictic acid, norstictic acid, constictic acid
*Pseudevernia furfuracea*	Lecanoromycetes, Lecanorales	NCBI NKYQ00000000.1	yes	37.8 Mb	46	1,178,799	8842	PFUR	atranorin, olivetoric acid, physodic acid
*Ramalina peruviana*	Lecanoromycetes, Lecanorales	NCBI MSTJ00000000.1	no	27.0 Mb	1657	40,431	9338	RPER	sekikaic acid complex
*Umbilicaria hispanica*	Lecanoromycetes, Umbilicariales	NCBI PKMA00000000.1	yes	41.2 Mb	1619	145,035	8488	LHIS	gyrophoric acid, lecanoric acid, umbilicaric acid, skyrin
*Umbilicaria muehlenbergii*	Lecanoromycetes, Umbilicariales	NCBI JFDN00000000.1	no	34.8 Mb	7	7,009,248	8968	UMUE	gyrophoric acid
*Umbilicaria pustulata*	Lecanoromycetes, Umbilicariales	NCBI FWEW00000000.1	yes	39.2 Mb	3891	104,297	8268	LPUS	gyrophoric acid, lecanoric acid, hiascinic acid, skyrin
*Xanthoria parietina*	Lecanoromycetes, Teloschistales	JGI Xanpa2 v1.1	yes	31.9 Mb	39	1,731,186	11,065	XPAR	physcion, parietinic acid, teloschistin, emodin

^1^ NCBI: National Center for Biotechnology Information; JGI: DOE Joint Genome Institute.

**Table 2 molecules-24-00203-t002:** Number of biosynthetic gene clusters and main families of secondary metabolite genes found in the genomes of lichen-forming fungi.

Species	Abbreviation	Number of Clusters	Type I NR-PKS	Type I R-PKS	Type I PKS	Type III PKS	Hybrid PKS-NRPS	NRPS	Terpene Synthases	Total KS Sequences for Phylogeny	Complete PKS (KS + AT + ACP)
*Cladonia grayi*	CGRA	51	8	17	1	3	1	2	5	27	15
*Cladonia macilenta*	CMAC	52	11	23	-	2	4	2	5	38	25
*Cladonia metacorallifera*	CMET	59	13	26	1	2	2	2	8	42	29
*Cladonia uncialis*	CUNC	59	10	25	1	2	3	1	8	39	25
*Endocarpon pusillum* [8] (Park et al.)	EPUP	31	4	9	-	1	2	2	6	15	9
*Endocarpon pusillum* [9] (Wang et al.)	EPUW	31	5	12	-	1	2	1	6	19	12
*Evernia prunastri*	EPRU	80	9	29	1	2	4	4	13	43	30
*Gyalolechia flavorubescens*	GFLA	37	7	12	-	1	1	3	6	20	10
*Lobaria pulmonaria*	LPUL	77	8	28	-	-	4	9	16	40	22
*Pseudevernia furfuracea*	PFUR	51	5	23	-	2	3	4	5	31	17
*Ramalina peruviana*	RPER	47	9	18	3	1	1	3	2	31	15
*Umbilicaria hispanica*	LHIS	28	7	10	1	1	-	-	6	18	14
*Umbilicaria muehlenbergii*	UMUE	31	5	15	-	1	-	-	4	20	17
*Umbilicaria pustulata*	LPUS	27	6	9	-	1	1	-	6	16	13
*Xanthoria parietina*	XPAR	51	5	18	1	1	2	5	7	26	13
Sum		712	112	274	9	25	30	38	125	425	266

NR-PKS = Non-reducing type I polyketide synthase; R-PKS = Reducing type I polyketide synthase; NRPS = Non-ribosomal peptide synthetase. KS = Ketosynthase; AT = Acyltransferase; ACP = Acyl carrier protein.

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
