# Peer review of "Biosynthetic Gene Content of the ‘Perfume Lichens’ Evernia prunastri and Pseudevernia furfuracea"

_molecules, 2019, doi:10.3390/molecules24010203_

Reviewer 1 Report

Quality analysis of data on gene clusters encoding secondary metabolism in lichen fungi, largely based on own genome sequences. Well written with good presentations.

Only a few comments and suggestions:

line 2: can omit “a”

l. 23-24: “...includes 15 genomes of lichenized fungi and all fungal...”

l. 28-29: “...facilitate heterologous...”

l. 40: “...represent complex...”

l. 42-43: “High-throughput sequencing technologies have revealed...”

l. 58: “...thousands of tons...”

l. 62: “synthetase (PKS)”

l. 65: “...subdivided into...”

l. 80: “...small fraction has been investigated experimentally to couple[connect] genes...”

l. 83: “...profile, hint at a plethora...”

l. 103: “encoded in the same...”

l. 113: “oxidation” (singular)

l. 121: “...phylogenetics of the ...”

l. 131: “...and thereby provide the most ...”

l. 171: “...together with basic genome ...”

l. 177: “.. metabolite enzymes identified ...”

l. 182: “It has been reported ...”

l. 184: “The genomes of the lichen-forming fungi ...”

l. 192: “… predicted secondary metabolite gene clusters ...”

l. 203: It would help if the term “Hybrid” would stand out in some way.

l. 207: “… a high number of natural products has been ...”  (verb refers to “a number”)

l. 212: “A” (end of line)

l. 220: “… contained an orthologous ...”

l. 223: “Four NR-PKS clusters show a core ...”

Figure 3: Can you comment on the significance of the homology of coding sequences to what appear to be non-coding sequences, eg. in A ?

Also, in the legend there is no explanation of the bolded numbers. It would also help here and in other figures if the differentiation of bold vs. non-bold was increased.

l. 260: “… PKS-NRPS and a NRPS ...”

l. 262-263: CONSIDER deleting sentence “Both core … … in E. prunastri.”

l. 287: “… as an outgroup. “

l. 288: “… also included partial ...”

l. 293: “… without an AT ...”    “… without an AT domain.”

l. 295: “… shows support for a clade ...”

l. 312: “have members within...”

l. 351: “… for developing heterologous ...”

p { margin-bottom: 0.1in; line-height: 115%; }

Reviewer 2 Report

This is a very nice study demonstrating the diversity of secondary metabolites in selected (and economically rather important!) lichenized Ascomycota. The methods used are state of the art and the paper is very well written. I just found some minor errors in the text that the authors can easily correct (see attached anotated pdf).

The only minor point of criticism is the selection of the species that the authors used for comparison with their own data. Of course, the mere numbers of genes and gene clusters do not tell anything because many of these clusters may in the end be shown to encode for rather uninteresting molecules like rare fatty acids and diketopiperazines, and the larger clusters are normally much more interesting. Howewer, the authors have definitely omitted some species in the Sordariomycetes that are rather rich in secondary metabolites as also reflected by the organisation of their genomes.  I have mentioned the species that i have missed in an annotation.

The paper, in any case, is almost ready to pubilsh and will definitely be interesting for a broad readership involving several scientific communities. I would like to congratuate the authors to their fine work!
